# The Validity and Feasibility of Utilizing the Photo-Assisted Dietary Intake Assessment among College Students and Elderly Individuals in China

**DOI:** 10.3390/nu16020211

**Published:** 2024-01-09

**Authors:** Rui Fan, Qianqian Chen, Lixia Song, Shuyue Wang, Mei You, Meng Cai, Xinping Wang, Yong Li, Meihong Xu

**Affiliations:** 1Department of Nutrition and Food Hygiene, School of Public Health, Peking University, Beijing 100191, China; fanruirf@bjmu.edu.cn (R.F.); cqq27170213@163.com (Q.C.); 1710306221@pku.edu.cn (L.S.); wangshuyue16@163.com (S.W.); 1610306135@pku.edu.cn (M.Y.); 1810306215@pku.edu.cn (M.C.); xinpingwang02@163.com (X.W.); liyong@bjmu.edu.cn (Y.L.); 2Beijing Key Laboratory of Toxicological Research and Risk Assessment for Food Safety, Peking University, Beijing 100191, China

**Keywords:** dietary assessment, Chinese elderly individuals, college students, food weight, photo-assisted dietary intake assessment (PAD)

## Abstract

Dietary assessments hold significant importance within the field of public health. However, the current methods employed for dietary assessments face certain limitations and challenges that necessitate improvement. The aim of our study was to develop a reliable and practical dietary assessment tool known as photo-assisted dietary intake assessment (PAD). In order to evaluate its validity, we conducted an analysis on a sample of 71 college students’ dinners at a buffet in a canteen. We compared estimates of food weights obtained through the 24-h recall (24 HR) or PAD method with those obtained through the weighing method; we also evaluated the feasibility of PAD for recording dinner intakes among a sample of college students (n = 76) and elderly individuals (n = 121). In addition, we successfully identified the dietary factors that have a significant impact on the bias observed in weight estimation. The findings of the study indicated that the PAD method exhibited a higher level of consistency with the weighing method compared to the 24 HR method. The discrepancy in D% values between cereals (14.28% vs. 40.59%, *P* < 0.05), vegetables (17.67% vs. 44.44%, *P* < 0.05), and meats (14.29% vs. 33.33%, *P* < 0.05) was clearly apparent. Moreover, a significant proportion of the food mass value acquired through the PAD method fell within the limits of agreement (LOAs), in closer proximity to the central horizontal line. Furthermore, vegetables, cereals, eggs, and meats, for which the primary importance lies in accuracy, exhibited a considerably higher bias with the 24 HR method compared to the PAD method (*P* < 0.05), implying that the PAD method has the potential to mitigate the quality bias associated with these food items in the 24 HR method. Additionally, the PAD method was well received and easily implemented by the college students and elderly individuals. In conclusion, the PAD method demonstrates a considerable level of accuracy and feasibility as a dietary assessment method that can be effectively employed across diverse populations.

## 1. Introduction

The significance of diet quality to human health is paramount, as a well-balanced dietary intake plays a crucial role in facilitating normal growth and development, maintaining overall health and well-being, and mitigating the risk of illness [1]. Accurate dietary assessments are imperative for determining one’s dietary and nutritional status, as it enables a comprehensive understanding of the relationship between diet and various health outcomes. This understanding, in turn, facilitates the formulation of effective public health policies and interventions [2]. The three primary methods for dietary assessments, dietary records (DR), 24-h dietary recalls (24 HR), and food frequency questionnaires (FFQ), are widely used to capture information on the consumption of various foods and nutrients through self-reported data [3]. For the assessment of dietary behavior and patterns, FFQ proves to be more suitable than DR and 24 HR [4]. Compared to 24 HR and DR, the FFQ method is more cost-effective and less intrusive, and it takes less time for participants [5]. Furthermore, these retrospective methods are susceptible to measurement errors, which involve the deliberate or unconscious omission of consumed foods [6]. Studies have shown that the correlations between food frequency questionnaires and the standards against which they have been compared typically range from 0.5 to 0.67, which has indicated that approximately 75–64% of the estimated variance is due to error [5]. Of course, the relatively accurate method has been the weighing food method, which has been widely recognized as the more precise approach to dietary assessments, and it has emerged as the preferred method. Even so, this method is highly expensive, and its substantial burden, including time and manpower, contributes to low participation rates and high attrition rates [7]. The ongoing use of traditional and classical methods to evaluate diets in scientific environments requires a thorough analysis and evaluation [8,9]. As a result, it is imperative to promptly create accurate and feasible tools for assessing diets that can effectively address the limitations and restrictions associated with these conventional approaches.

Indeed, the main obstacle in dietary assessment concerns the identification and quantification of ingested foods, a task that is even more intricate within the framework of Chinese food culture, given the extensive variety of food types accessible and the diverse assortment of cooking techniques employed [10,11]. Currently, there is a growing trend of utilizing food photography or electronic images, wearable cameras, and diverse online tools as emerging innovations. These innovations have demonstrated their potential to enhance participants’ compliance by alleviating the burden of recording and improving the accuracy of recorded data [12,13,14]. Furthermore, a considerable number of these techniques facilitate the real-time or near-real-time transmission of food image data to researchers or clinicians for analysis and guidance [15,16]. The current innovative assistive tool has exhibited its capacity to alleviate recall bias among participants and even eliminate the need for users to estimate portion sizes in specific scenarios. Notably, the image-assisted approach to estimating food portion sizes has exhibited a strong correlation with weighed portion sizes, with a deviation of less than 10% in the overestimation of image-based estimates compared to the weighing method [17,18]. However, this promising conclusion was derived from a study conducted on foreign populations. It is imperative to investigate the potential among Chinese individuals due to their distinct eating and cooking habits. Furthermore, an alternative technology-based food recording instrument, particularly one that is reliant on the internet, has been documented to underestimate micronutrient and food group consumption in comparison to conventional dietary evaluations [19]. Moreover, the reliability of web-based dietary assessments has been particularly restricted for the elderly population [19,20,21]. In the field of Chinese studies, there has been a scarcity of research conducted on image-/web-assisted methodologies. Ding conducted an assessment of a WeChat applet that employed an image-based dietary evaluation technique. The principal technological facet of this approach entailed converting the food’s area into its mass to facilitate quantification. Nevertheless, this methodology failed to consider the food’s dimensions, including its thickness and height, which could influence the accuracy of the results. Additionally, Ding’s research focused solely on pregnant women, overlooking the potential difficulties associated with implementing this approach among college students, who lack life experience, and the elderly, who may have diminished physical capabilities [22]. Therefore, in order to evaluate the feasibility and accuracy of the image-assisted method for dietary assessments, it is imperative to select diverse settings and populations.

Given the distinctiveness of Chinese food culture, the intricacy of Chinese culinary practices, and the wide range of food and dishes available, this study introduced a photo-assisted dietary intake assessment (PAD) method. The objective was to assess the accuracy of this method by comparing the food mass obtained through PAD with the gold-standard weighing method and the commonly used 24-h dietary recall. Additionally, both elderly individuals and college students were included in the study to evaluate the feasibility of PAD. The findings of this study have the potential to contribute to the advancement of dietary surveys and assessments, thereby enhancing public health and well-being.

## 2. Materials and Methods

### 2.1. Overview of the Photo-Assisted Dietary Intake Assessment (PAD)

The PAD comprises three parts: a food atlas, an information collection system, and a data processing system (Figure 1). The food atlas was prepared by researchers who underwent specialized training. Its purpose was to quantify mass through the conversion of food volume in specifically designed bowls into food mass. It included 150 images representing 70 different food items. Each food item was depicted in two images captured from two different angles (directly above and in front at a 45° angle). Each image depicted various food items presented in bowls of either half or one unit volume (i.e., 60 or 120 cm^3^). Basically, the bowls are structured as a set of five bowls, all possessing an identical base area of 120 cm^2^. However, their heights differ to cater to distinct functions, as depicted in Appendix A. These functions encompass the containment of staple foods (5 cm in height), vegetables (3 cm in height), meats (2 cm in height), and fruits or nuts (2 cm in height).

The establishment of the food atlas entailed a methodical procedure comprising five sequential steps whose specific information is elucidated in the Appendix A:(1)The collection of food items, including information on their processing and cooking methods;(2)The determination of the percentage of edible portions of foods and the raw/cooked ratio;(3)The determination of sizing references and portion sizes;(4)The documentation of food pictures;(5)The compilation of these food pictures to form the food atlas.

During the dietary assessment process, the participants utilized their smartphones to capture photos of their food intakes, which were subsequently transferred to the researchers through the WeChat software within the community. Once the dietary information was received by the researchers via WeChat, the task of assessing the dietary intakes was assigned to the dietary assessors. The assessors proceeded to identify and estimate the volume of each food item depicted in the images, subsequently converting the volume measurements into corresponding mass values, using the food atlas as a reference.

### 2.2. Validation of the PAD

#### 2.2.1. Participants

An observational study was undertaken to validate the PAD. A total of 76 college students, aged between 22 and 24 years, who communally ate at a canteen, were invited to participate in this study. The study was conducted during dinner time at the buffet stalls in Yuejin Canteen, located at the Peking University Health Science Center in Beijing, China. Prior to their participation, all the participants provided their informed consent for their inclusion in the study. However, as a result of insufficient weighing data for five students, only the data from a total of 71 students were utilized for the purpose of analysis. Peking University’s Biomedical Ethics Committee approved the protocol, which was conducted in accordance with the Declaration of Helsinki (IRB00001052-22121, 13 October 2022).

#### 2.2.2. Study Design

The selection of university cafeterias as this study’s research setting was based on their ability to offer a wide variety of foods from different regions in China, thus encompassing representative culinary offerings. To ensure the feasibility of the investigation and account for the diversity of meals, only one dinner option was chosen from the buffet stall. This dinner was served in a buffet style, and participants were provided with the same types of food, although variations in the quantity and selection of the items were offered to each individual.

The study design is depicted in Figure 2. Prior to the commencement of the observational study, the study protocol was elucidated to both the canteen staff and the participants. The participants were informed that they would be provided with a dinner served in specially designed bowls on the initial day, and they were instructed to capture photographs before and after consuming their meals. Subsequently, they were required to return on the following day to complete the 24-h recall dietary survey.

#### 2.2.3. Dinner Preparation and Food Weighing

A meal was prepared based on menus that had been communicated to the researchers according to the common processes. The dinner consisted of a total of 30 dishes, as specified in Appendix A. The primary ingredients used in these dishes encompassed rice, wheat, tomato, Chinese cabbage, mushroom, rape (scientific name: *Brassica napus* L.; Chinese name: you cai; a green leafy vegetable), pork, grass carp (Chinese name: cao yu), egg, chicken, ribs, potato, sweet potato, yam, and black fungus, among others.

Once the dinner had been prepared, the participants were required to carefully select their food portions and place them into clean, specially designed bowls (with a total of 5 bowls per participant). The researchers then proceeded to measure the mass of each food type. Subsequently, after the meal, any remaining food was reweighed by the researchers.

#### 2.2.4. The PAD Method

When the food was served, the various food items were arranged in distinct sections within specially designed bowls, ensuring that all the food items reached the same height. Subsequently, the participants captured photographs of the food from two perspectives: directly above and in front at a 45° angle. Following the meal, the same photographic method was employed to capture images of any remaining food. Ultimately, the food images were uploaded to the researchers via the WeChat applet, serving as records for analysis.

#### 2.2.5. The 24-H Recall Method

In light of the extensive utilization of 24 HR in dietary evaluations, the current study likewise employed 24 HR as a benchmark to evaluate the comparative precision of the PAD method. Well-qualified researchers conducted face-to-face interviews with the participants, collecting data through 24 HR on the second day. To facilitate the estimation of food portion sizes, food models were provided. The participants’ responses were documented on a dietary recall form, and subsequent to the interviews, the researchers diligently addressed any ambiguous information provided by the participants.

### 2.3. Application of the PAD Method

#### 2.3.1. Participants

This section of the survey involved the involvement of both college students and elders, who were selected as representatives of the target population. The college students included in this survey were identical to the population that had engaged in the validation trials expounded upon in Section 2.2.1. Ultimately, a total of 76 students were enlisted for the feasibility trial.

An observational study was undertaken to evaluate the utilization of the PAD among elderly individuals aged 60 to 70 years old. The study was conducted over a period spanning from March 2023 to June 2023, and it involved the recruitment of elderly residents from Chengdu City. In order to be included in the study, the participants had to meet the eligibility criteria of lacking any clinical diagnosis of infectious disease. Those who had limited cognitive abilities or were unable to use a smartphone were excluded from the study. All participants gave their informed consent prior to participating in the study, which was conducted in compliance with the Declaration of Helsinki and approved by the Biomedical Ethics Committee of Peking University (IRB00001052-21114, 17 November 2021). The final eligible data included the recordings of 121 elders’ dinners.

#### 2.3.2. Study Design

On the day preceding the commencement of the observational study, the participants were briefed on the procedures of the dietary survey, including the utilization of specially designed bowls and photography. Additionally, they were instructed to submit photographs of their dinners, taken both before and after consumption, via WeChat. Furthermore, the participants were required to return on the subsequent day to complete the 24-h dietary survey.

### 2.4. Feedback on the PAD Application

During the feedback sessions, a questionnaire was administered to the college students on the second day to assess their perceptions and attitudes towards the PAD. In response to these questions, the participants were asked to select one of five ordinal response options, including strongly agree, agree, neutral, disagree, and strongly disagree.

The participants were asked to respond to the following five statements regarding their perceptions: (1) It would be easy to remember to take a photo before meals, (2) it would be easy to remember to take a photo after meals again, (3) it would not be difficult to know what I intake, (4) I think I am willing to cooperate, and (5) the PAD/24 HR was easy to carry out [23].

In light of the challenges faced by the elderly population, the elders did not undertake a comprehensive evaluation of the questionnaire. These five inquiries could have been effectively addressed by soliciting their feedback on the photographs, such as assessing whether they had forgotten to capture images before or after meals, verifying whether the photographs met the specified criteria, and determining whether any complaints had been received. Based on the responses, the researchers categorized their responses into five dimensions in order to thoroughly assess the attributes of the PAD and 24 HR methods. According to the previous report [24], each dimension was assigned a rating on a 5-point scale with ×× = very poor, × = poor, O = fair, √ = good, and √√ = excellent.

### 2.5. Quality Control

Before commencing the study, all the researchers underwent a mandatory technical skill training program that lasted at least two days. Only those who achieved satisfactory performance in the program’s examination were deemed eligible to partake in the study. Furthermore, in order to uphold the integrity of the investigation, the researchers were segregated into three distinct working groups: dietary weighting, dietary recall survey, and dietary assessment. Each group functioned autonomously, with no intermingling of personnel or information among them.

### 2.6. Statistical Analysis

To ascertain the dependability of the survey instrument, the paired design difference-in-differences mean test can be employed in order to determine the appropriate sample size (Equation (1)):(1)$$n=\frac{{\left(\mu_{\alpha }+\mu_{\beta }\right)}^{2}\sigma^{2}}{\delta^{2}}$$
where α = 0.05, and β = 0.01; the standard deviation of the dietary energy intake among pregnant individuals was reported as 400 kcal/d in a previous study [25]. Additionally, the difference in dietary energy intake between the photographic method and the 24-h recall method among the adult population was found to be 269 kcal [26]. After performing calculations, it was determined that a minimum of 25 participants would be required, accounting for a non-compliance rate of 10%. However, in order to ensure sufficient data, and given relevant reports [16,26,27], a total of 71 participants were enrolled in this study.

We calculated and analyzed all the data using SPSS, version 20.0 (IBM Corp., Armonk, NY, USA). The differences were expressed as means and standard deviations (SDs) for normal distributions and as medians and quartiles (the first and third quartiles) for non-normal distributions. Statistical significance was defined as *P* = 0.05.

In accordance with the Chinese Balanced Dietary Pagoda and Chinese dietary habits, the weights of the food items belonging to the different groups were estimated with the 24 HR or PAD method, and the results were compared with the food items’ actual weights measured using the weighing method.

The relative difference (d) and absolute difference (D) were calculated as follows:Relative difference (d) = weight obtained with recall or PAD-actual weight(2)
Absolute difference (D) = |weight obtained with recall or PAD-actual weight|(3)
Percentage d(d%) = d/(actual weight) × 100(4)
Percentage D(D%) = D/(actual weight) × 100(5)

Paired *t*-tests were employed to compare the food groups between the 24 HR and PAD methods. Spearman correlation coefficients were utilized to assess the correlation between food weights obtained using the 24 HR or PAD method and the weighing method. Subsequently, a Bland–Altman analysis was employed to evaluate the agreement between the distribution of the differences in food weights obtained from the 24 HR or PAD method and the weighing method. Calculations of the 95% confidence interval of the differences and the 95% limit of agreement (LOA) were used to observe the dispersion trend of the difference.

Moreover, to assess the potential bias in the contribution of nine food items between the 24 HR and PDA methods, a principal component analysis (PCA) was conducted to reduce the dimensionality of the food items while preserving the majority of the variance in the data. During the PCA analysis, a correlation matrix was derived through varimax rotation, and only eigenvalues greater than 1 were considered to determine the number of components. The scree test was employed to illustrate the extent of variation accounted for by each of the primary components [28].

## 3. Results

### 3.1. Comparison between the Weighing and PAD Methods in Terms of Food Weights

Figure 3 demonstrates that the weights identified through the weighing method exceeded those identified via the PAD method for the majority of food items, encompassing cereals, tubers, soybeans and their derivatives, and starchy products. Specifically, the weights of these four food categories were lower when measured using the PAD method compared to the standardized weights, a trend observed for both male and female subjects. Notably, the mass of tubers obtained through the PAD method exhibited a statistically significant decrease in both the male and female populations (*P* < 0.05). Moreover, the mass of starchy products identified through the PAD process exhibited a notable decrease compared to the established weight for males. Additionally, the weight of eggs identified via PAD demonstrated no significant difference when compared across different genders (*P* > 0.05). For males, the mass of meats identified with PAD was found to be lower than the standard weight, whereas for females, the opposite trend was observed. The weight of mushrooms identified through PAD was found to be significantly greater than the standard weight for males. Conversely, for females, the weight of mushrooms was observed to be lower than the standard weight. Similarly, the weight of blood curds and vegetables obtained through PAD was lower than the standard weight for males but exhibited a comparable mass to the standard value for females.

According to the data presented in Table 1, the weights of various food items (including cereals, tubers, soybeans and related products, vegetables, and starch products) identified through the use of PAD were found to be lower than the standardized weights for both males and females. This finding aligns with the results depicted in Figure 3. The findings of the blood curd analysis aligned with the outcomes presented in Figure 3. Specifically, the weights identified using PAD were lower than the standardized values for males, whereas for females, the weights were comparable to the standardized values. Conversely, the results for the mass of meats and mushrooms exhibited contrasting patterns between males and females. Males underestimated the weight of meats compared to the standard weight, while females overestimated the weights, surpassing the standard value. Furthermore, the accuracy of meat weight estimations by females was significantly superior to that of males, as indicated by a statistically significant difference (*P* < 0.05). Similarly, there was a significant disparity in the estimated mass of tubers between male and female individuals (*P* < 0.05). In summary, while there was some variation in the estimated weight of foods identified through PAD compared to the actual weights, the majority of these differences were not statistically significant. Furthermore, the differences observed between genders were found to be insignificant when utilizing the PAD method.

### 3.2. Comparison and Correlation Analysis among Different Methods

#### 3.2.1. The Analysis of Food Weights Estimated Using the 24 HR or PAD Method Versus the Actual Food Weights

The results depicted in Figure 4 demonstrated variations in food weights across the three dietary survey methods (weighing, 24 HR, and the PAD method). For cereals, vegetables, starch products, mushrooms, and blood curds, the weights assessed via the three methods achieved the same results, i.e., weighing > PAD > 24 HR, which partly indicated that the accuracy of the PAD method fell between that of the weighing method and that of the 24 HR method for these five foods. Additionally, there was an obvious difference in the mass of cereals among the three methods (*P* < 0.05), suggesting that the accuracy of the PAD method in weight assessment was superior to that of the 24 HR method. In contrast, the weights of tubers, soybeans and related products, and eggs exhibited consistency across the three methods., i.e., weighing > 24 HR > PAD, which suggests that the precision of the evaluated weights using PAD was marginally lower compared to those using the 24 HR method for the aforementioned three food items. In terms of meat, it was evident that the actual mass was significantly lower than that determined via the PAD method but noticeably higher than that determined via the 24 HR method. Furthermore, the three methods exhibited a significant difference in accuracy (*P* < 0.05). Overall, there were no distinct advantages or disadvantages between the 24 HR and PAD methods for most food items except for cereals; cereals just happened to be the most frequently consumed food, which means that the PAD for the weight estimation of foods with more consumption exhibited a certain advantage.

According to the data presented in Table 2, the weights of various food items (such as cereals, tubers, soybeans and related products, vegetables, and starch products) identified through 24 HR and PAD methods were found to be lower than the actual weights. Specifically, the weights of eggs and blood curd identified through the 24 HR method were also lower than the actual weights, whereas the weights estimated by the PAD method were consistent with the actual weights. Conversely, the weights of mushrooms and meats identified through the 24 HR method were lower than the actual weights, while those identified through the PAD method were higher and closer to the actual weights. Furthermore, it was evident that the absolute deviations of the PAD method were consistently smaller than those of the 24 HR method for all food categories. Notably, cereals, soybeans and related products, vegetables, mushrooms, meats, and eggs exhibited significantly smaller absolute deviations with the PAD method compared to the 24 HR method. This observation suggests that the PAD method, particularly for these six food types that are representative of daily dietary intakes, is significantly more precise than the 24 HR method.

#### 3.2.2. The correlation and Bland–Altman Analysis among Different Methods

A correlation analysis was conducted, and the findings are presented in Table 3. The Spearman correlation coefficients between the actual weights (using the weighting method) and the weights according to 24 HR varied from 0.017 for starch products to 0.776 for tubers. Significant correlations (*P* < 0.05) were observed for cereals, tubers, vegetables, meats, and eggs. Similarly, the Spearman correlation coefficients between the actual weights and the weights according to the PAD method ranged from 0.743 for tubers to 0. 944 for meats, and the correlations of all food items were statistically significant (*P* < 0.05). These findings indicate that the weights derived from the PAD method exhibited a higher degree of proximity to the true weights compared to the 24 HR method.

To evaluate the concordance between the mass determined through the weighing method and the estimations via the 24 HR or PAD method, a Bland–Altman analysis was conducted. The results are presented in Table 4. In general, the mean differences in the food mass estimated using the 24 HR method exhibited a greater deviation from zero compared to the weighing method, as well as wider 95% confidence intervals when compared to the PAD method. The range between the lower and upper LOAs for the masses identified through the 24 HR method was also wider than for that obtained through the PAD method. 

Additionally, Bland–Altman plots were employed to illustrate the correlation between the average and the disparity in the mass intake identified through both the weighing technique and the 24 HR or PAD method (Figure 5). According to the data presented in Figure 5, the x-axis denotes the average total intake mass identified through the weighing method, as well as that identified through the 24 HR or PAD method, while the y-axis represents the discrepancy in mass between these three methods. Compared with 24 HR, an obvious good agreement in the mass estimated using PAD and the actual mass was shown for meats, mushrooms, eggs, and soybeans, for which no points exceeded 95% LOAs, and the points were near the central horizontal line, while the poor agreement of the mass estimated with 24 HR and the actual mass was shown for mushrooms and eggs, reflecting the exceeding of 95% LOA in 10% of the points. The cereals and vegetables assessed using the PAD method exhibited a greater proximity to the 0 horizontal line compared to those assessed using the 24 HR method, although some points exceeded 95% LOAs with the PAD and 24 HR methods. The results indicate that the PAD method may present specific benefits compared to the 24 HR method regarding weight evaluations for particular food items.

#### 3.2.3. The Principal Component Analysis

T Do assess the influence of bias in the weights used in the 24 HR or PAD methods, the D% values of all food items were utilized in the adoption of principal component analysis (PCA). The decision to employ PCA was based on the stronger linear correlation observed between the food weights evaluated by 24 HR or PAD and their standardized weights, as well as the statistical significance indicated by a *P*-value of less than 0.001. Subsequently, cereals, tubers, vegetables, meats, and eggs (excluding starch products, mushrooms, blood curd, and soybeans due to a limited sample size) were subjected to further analysis using PCA. Overall, it is considered acceptable to have KMO values above 0.50 and p values of 0.05 for Bartlett’s sphericity test.

The PCA conducted in this study revealed that multiple factors contribute to the food weight bias observed in the 24 HR or PAD methods. Figure 6A demonstrates the presence of two components with an eigenvalue greater than 1, which collectively account for 81.34% of the bias observed in the 24 HR method. Similarly, Figure 6B illustrates the presence of two components with an eigenvalue greater than 1, explaining 69.21% of the bias observed in the PAD method.

According to the statistical requirements, it was suggested that commonalities (≥0.5) and component loadings (≥0.7) were fulfilled [29]. The analysis of Table 5 revealed that cereals, vegetables, and meats from component 1, along with tubers from component 2, effectively accounted for the bias in weights assessed using PAD. Similarly, cereals, vegetables, eggs, and meats from component 1, along with tubers from component 2, adequately explained the bias in weights assessed using 24 HR. Above all, the primary focus of importance lay in the impact of accuracy on the estimated mass of cereals, tubers, vegetables, eggs, and meats in 24 HR and cereals, tubers, vegetables, and meats in the PAD method.

After the substantial influence of cereals, tubers, vegetables, meats, and eggs on the accuracy of 24 HR or PAD was recognized, a comparison was conducted using these food records, and its outcomes are presented in Table 6. There existed a discernible level of bias in the assessment of food types and weights when comparing the 24 HR and PAD methods. However, this bias was not statistically significant for food items (*P* > 0.05) except meats with less than 100% edible portions (*P* < 0.05). There was a significant difference in weight estimation between the 24-h dietary recall (24 HR) and photographic food atlas (PAD) methods for tubers, vegetables (root and stem, melon and solanaceous products, and flowers and leafy products), cereals, and meats with 100% edible portions. This disparity was evident in the proportion of deviations exceeding 50% in the weight estimations with statistical significance (*P* < 0.05). The study demonstrated a significant increase in accuracy for cereals, eggs, meats, and vegetables when utilizing PAD (*P* < 0.05), which implied that the PAD method has the potential to mitigate the quality bias associated with these food items in the 24 HR method. Meanwhile, the proportion of deviations exceeding 50% in estimating tuber weights using the 24 HR method was notably lower compared to that of the PAD method (*P* < 0.05).

### 3.3. The Application of Recall and the PAD Method to Different Populations

The PAD and 24 HR methodologies were utilized to assess the dietary surveys of college students in cafeterias and elderly individuals at home, respectively. The findings are presented in Figure 7. In relation to the PAD approach, it was noted that 21% of the elderly participants and 17% of the college students either exhibited complete amnesia regarding their dietary intakes or provided entirely inaccurate information when using the 24 HR method. This difference was not statistically significant between the two populations (*P* > 0.05). Furthermore, the male-to-female ratio was approximately 1:1 in both groups, and there was no significant variation between the two populations in terms of gender (*P* > 0.05). This finding suggests to some degree that the PAD method is appropriate for conducting dietary surveys among college students who consume canteen meals and elderly individuals residing at home due to its reducing recall bias. Furthermore, it was evident that both elders and students perceived the PAD method to be less burdensome and challenging compared to the 24 HR method. Additionally, a higher level of cooperation was observed with the PAD method. Consequently, it can be concluded that PAD exhibits a clear advantage in terms of its applicability, feasibility, and acceptance in dietary assessment.

## 4. Discussion

The accuracy of dietary assessments and surveys holds significant importance in the field of public health. In order to address the limitations of current dietary survey methodologies, a novel dietary assessment tool, PAD, was introduced in this study. The validity of PAD was assessed by comparing it with the weighing method and 24 HR among college students. Additionally, the utilization of PAD was evaluated among college students and elderly individuals who were residing communally and given access to canteens and rent-free homes, respectively. The results indicated a lower deviation and closer proximity for PAD versus the weighing method. Additionally, the participants’ willingness and responses were evaluated. Based on these findings, it can be concluded that PAD is a viable approach to assessing dietary quality in China.

It is important to determine whether a dietary assessment method is valid when developing a new one. Validity refers to the degree to which an assessment method captures true dietary intakes [30]. This validation necessitates a comparison with a recognized benchmark method, commonly referred to as the “gold standard” [22]. Consequently, the weighing method was chosen for the standard weights in the present study. Additionally, the 24 HR method, which possesses a distinct error structure, was selected to evaluate the relative validity of PAD [31]. The findings of our study indicated that the weights of most foods, including cereals and vegetables, were lower with PAD than the standardized weights. In terms of food mass, previous studies using image-assisted and weighing methods have yielded inconsistent results, with some reporting larger or smaller masses than the actual values, so the current results were inconsistent with these reports [22]. The potential factors contributing to this phenomenon could be attributed to variations in participants, intricate dietary patterns, and diverse eating behaviors. Nevertheless, our present findings, indicating that the mass estimated using PAD was lower than the actual mass, align with previous studies conducted on Chinese populations [18,22]. Furthermore, the current study revealed that the median deviation in food weights between PAD and the actual mass (ranging from 0% to 33.33%) was greater compared to previous findings, which encompassed the comparison of image assistance coupled with manual portion size estimation method vs. weighing, and image assistance coupled with automatic portion size estimation vs. weighing [18,32,33,34]. It is unsurprising that employing statistical pattern recognition techniques for the automated estimation of food mass from images may yield greater accuracy compared to manual estimations. However, certain experts have maintained reservations regarding the accuracy of automated image analysis in estimating food intakes [24]. Our findings indicated a larger difference deviation in food weights compared to previous studies, which can be partially attributed to two factors. Firstly, China’s distinct eating culture and habits contribute to the inclusion of multiple ingredients in dishes and non-standard portion sizes. Secondly, our research was conducted in a communal dining hall for college students, providing a setting closer to real-life dining scenarios than controlled laboratory environments [24]. The food mass difference deviation between PAD and the weighing method for Chinese participants was found to be similar to the results of Ding’s study [22]. A good finding is that our study also revealed that certain food items, such as vegetables and meats, exhibited smaller deviations between PAD and the weighing method compared to the previous report (14.35% vs. 30.13% and 14.29% vs. 28.34%, respectively) [22]. In addition, the findings of our study indicate that the PDA method yielded less variation in estimated food mass (including for cereals, soybeans and related products, vegetables, mushrooms, algae, meats, and eggs) compared to the 24 HR method, which is consistent with a previous report [22]. The limitations of self-reporting methods, including 24 HR, and the sources of error associated with them encompass several factors. These include the unintentional under-reporting of consumed foods due to forgetfulness, the deliberate under-reporting of foods with unfavorable health connotations, such as those high in fat or sugar, the deliberate over-reporting of foods perceived as healthy, such as fruits and vegetables, and errors in estimating portion sizes [35,36]. The use of PAD can alleviate the user burden (as shown in Figure 7) and eliminate the necessity for users to make subjective estimations of portion size, as this task was performed by our trained assessors. Consequently, the accuracy of food mass estimation was enhanced. This finding is supported by the Spearman correlation coefficients between the actual food weights and the weights determined through the PDA method, as well as the Bland–Altman analyses presented in Table 3 and Table 4 and Figure 5. These results demonstrate that the majority of food weights identified through the PAD method are more precise than those identified through 24 HR.

Furthermore, our study revealed that the prominent impacts of 24 HR on bias in food mass were observed in cereals, tubers, vegetables, meats, and eggs, and PAD exhibited biases primarily in cereals, meats, tubers, and vegetables, with the sole exception being eggs. Indeed, eggs’ mass estimated via the PAD method exhibited closer proximity to the actual value compared to the 24 HR method (Table 3). One possible explanation could be attributed to the Chinese food culture, in which eggs are traditionally served whole and consumed directly, thus leading to a tendency to omit them. Eggs were found to be present in various culinary preparations, including dishes, soups, and even a slurry, as indicated by the food atlas based on the PAD estimation. These findings align with the results presented in Table 5, which demonstrate fewer instances of omitting food, and a difference deviation of more than 50% was clearly observed for the PAD method, rather than the 24 HR method (*P* < 0.05). In general, the utilization of PAD yielded higher levels of accuracy in estimating food mass compared to the 24 HR method. This was evident in the reduced occurrence of incorrect food item identification for meats with less than 100% edible content (*P* < 0.05), as well as the occurrence of deviations exceeding 50% for cereals, eggs, meats, and vegetables (*P* < 0.05). The findings highlight the significance of accuracy in the PAD method, particularly in relation to meats, eggs, cereals, and vegetables, and suggest that the implementation of PAD can effectively address these accuracy concerns in 24 HR. Ultimately, the accuracy of PAD in estimating food mass for dietary assessments was demonstrated.

In the case of certain food items, such as meat, a subset of participants acknowledged the potential underestimation of their quantities [37]. In the employed PAD method, the identification and weighing of food were conducted by researchers, rather than participants, thereby minimizing technical bias, enhancing the precision of weight data, and mitigating the influence of weighing records on the data collected through 24 HR. Furthermore, a notable absence of correlation exists between the visual perception of the appearance of completely consumable food and the corresponding food’s weight [38]. Consequently, with the 24-h dietary recall method, participants were unable to precisely recollect and assess the weights of food items that contained inedible components, such as chicken legs, resulting in a substantial discrepancy between the recalled values and the actual values. With the PAD method, the dietary assessors utilized the food atlas to estimate the inedible portions of foods. This involved fixing the percentage of inedible foods based on the average proportion from the majority of meats, such as chicken/duck legs, ribs, fish, and shrimp. However, it should be noted that the edible proportions of different food parts varied, and using average proportion data instead of actual data may introduce some degree of error. Additionally, it is worth mentioning that the good acceptance and implementation of the PAD method were observed for both students and the elderly population, as evidenced by the questionnaire and the feedback received (Figure 6B).

Tubers exhibited an exceptional profile, wherein the bias value for tuber deviation was observed to be smaller when employing the 24 HR method as opposed to the PAD method. This discrepancy may be attributed to Chinese eating habits, specifically the consumption of tubers such as sweet potato, yam, and potato, which are commonly considered staple foods and are typically served whole for direct consumption, rather than being placed in a container for photographing, which often leads to the omission of captured images, while eating the whole food makes it easier to recall and convert the measurements into its mass. The exclusion of dairy products and beverages from this study may perplex readers, as these items typically come in pre-packaged forms with specified net contents. Estimating the consumption of beverages is generally simpler than estimating the intake of food. For the estimation of tuber mass, there was an insignificant difference observed in the weights obtained through three methods (see Figure 3 and Table 2). However, it is worth noting that the percentage of derivation exceeding 50% was higher with the PAD method compared to the 24 HR method, which could be attributed to the smaller sample size.

Despite the presence of exceptions, the PAD method exhibited a multitude of advantages. On one hand, the present study aligned more closely with the actuality of Chinese diets by considering the distinctive features of Chinese dietetic culture. This study encompassed a diverse range of culinary items, encompassing starchy products such as fish balls, fish bean curd, vermicelli, sausage with starch, etc., as well as blood curd (specifically duck blood). These particular dishes have not been previously documented. Furthermore, the weights of all the food items were approximated by considering mixed dishes, i.e., photographs capturing the entirety of the mixed dish and subsequently recording and estimating the weight of each ingredient within the mixed dish. This estimation process was conducted using a substantial amount of data and the creation of a comprehensive food atlas. The weighing method necessitates the individual weighing of each ingredient in a dish, demanding a significant level of dexterity and expertise, thus resulting in a more intricate and time-intensive process compared to the PAD method. Furthermore, a notable distinction between the PAD method and previously reported image-assisted methods lies in their approaches to converting food weights. Specifically, the PAD method employs a graduated bowl to serve a mixed dish, ensuring an accuracy of 0.1 cm. This allows for the recognition and conversion of the volume occupied by each food into its corresponding mass. Consequently, the PAD method effectively mitigates the loss of mass attributed to the height or thickness of a food, a limitation observed for prior methods [10,22,34]. This advantage is particularly applicable to estimating the weights of ingredients that exhibit diverse shapes, such as chunks, bars, dice, and so forth. However, it should be noted that the estimated volume of foods with irregular shapes may introduce certain inaccuracies that cannot be mitigated using the existing assessment methodologies. The inclusion of dish images serves as a valuable aid in identifying the cooking technique employed, thereby influencing the alteration of food mass, particularly in the context of Chinese cuisine [38,39]. China uses various cooking methods, such as cold eggplant and roasted eggplant, each of which has distinct effects on the mass of the vegetable. Cold eggplant is typically steamed, resulting in a significant loss of water during the steaming process, leading to a raw/cooked ratio of approximately 50%. On the other hand, roasted eggplant is deep-fried and coated with a batter (e.g., flour and egg wash) prior to frying, resulting in minimal water loss with an increase in the weight of the eggplant [40]. Consequently, the different cooking methods pose challenges in accurately estimating the weight of the eggplant. The utilization of the proposed method can enhance the precision of weight estimations based on the cooking technique employed in this research. Consequently, the employment of PAD can effectively guarantee the accuracy of dietary assessments to the greatest extent possible. On the other hand, this study encompassed a diverse range of participants in terms of ages and meal patterns, including individuals who dined at group canteens and prepared meals at home, thus representing a wide spectrum of dining styles [41,42]. Moreover, the study specifically targeted college students with limited life and cooking experience, as well as elderly individuals with declining physiology, both of whom require special attention and may exhibit reduced accuracy [18,43]. Consequently, the utilization of the PAD method in this study effectively demonstrated its suitability and practical implementation in dietary assessments.

Another limitation that should be elucidated pertains to the exclusion of certain types of foods, specifically condiments. Condiments pose challenges when employing the PAD method for analysis due to their minimal mass and the inherent difficulty in estimating their quantities even using alternative methodologies. In the evaluation of the PAD accuracy, fruits were not involved; the reason was partly due to the forbidding of raw food ingredients in the group canteens for food safety during the COVID-19 period. In addition, fruits are always eaten directly as a whole or a half, rather than put in containers, and they are eaten during two meal intervals, rather than during meal time; this consumption pattern was just observed in this study among the elderly population. Furthermore, this study solely examined the comparative accuracy of dinner consumption. It was observed that numerous college students who frequently skipped breakfast or lunch due to waking up late or missing lunchtime often resorted to irregular eating patterns. Although dinner alone does not encompass the entirety of a day’s dietary intake, it holds significant importance as the primary meal in Chinese culture. Consequently, dinner can be considered somewhat representative in validating the PAD method.

## 5. Conclusions

This study has presented evidence indicating that the PDA method for the weight estimation of diverse food items demonstrates consistency with the weighing method, yielding a closer approximation to the actual value of dietary mass compared to the 24 HR method. Additionally, the PDA method demonstrates remarkable feasibility in terms of operationalization, application, and acceptability compared to the 24 HR method. However, to enhance the generalizability of the findings and determine the most suitable method for specific settings and target groups, larger sample sizes are necessary. Furthermore, the inclusion of nutritional dietary biomarkers would be beneficial in reflecting habitual dietary intakes.

## Figures and Tables

**Figure 1 nutrients-16-00211-f001:**
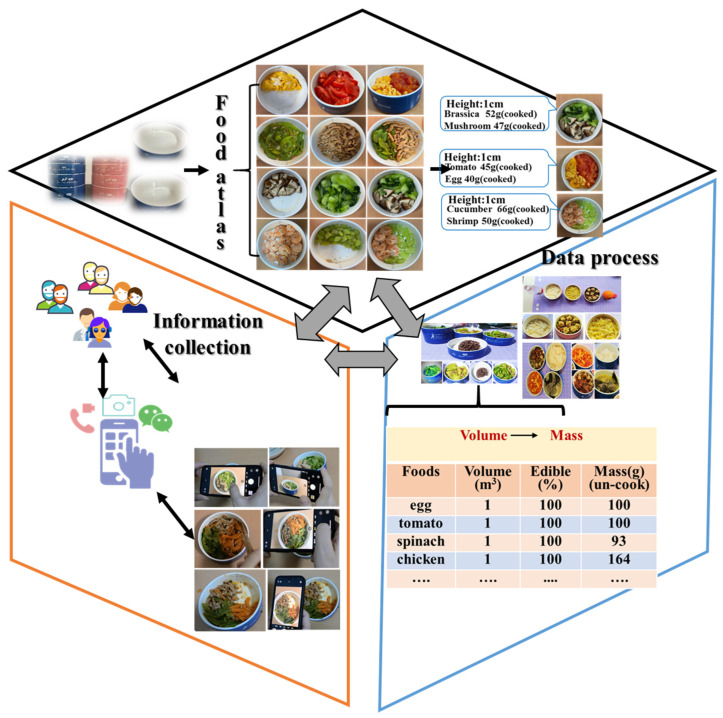
Overview of the photo-assisted dietary intake assessment (PAD).

**Figure 2 nutrients-16-00211-f002:**
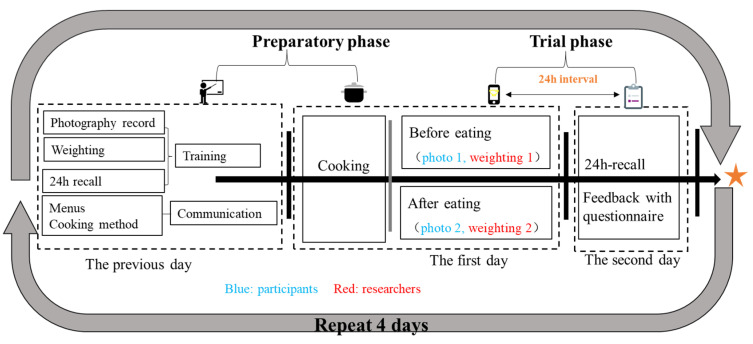
The design of the dietary assessment study. In the preparatory phase, menus had been communicated to the researchers according to the common processes, and the photography record operation was training to the participants, and weighting and 24 HR were training for the researchers. The trial phase lasted two days, on the first day, the participants took the photo for the foods before and after meals, and researchers weighted the mass of all food items. On the second day, the participants were received the 24 recall and the feedback with questionnaire.

**Figure 3 nutrients-16-00211-f003:**
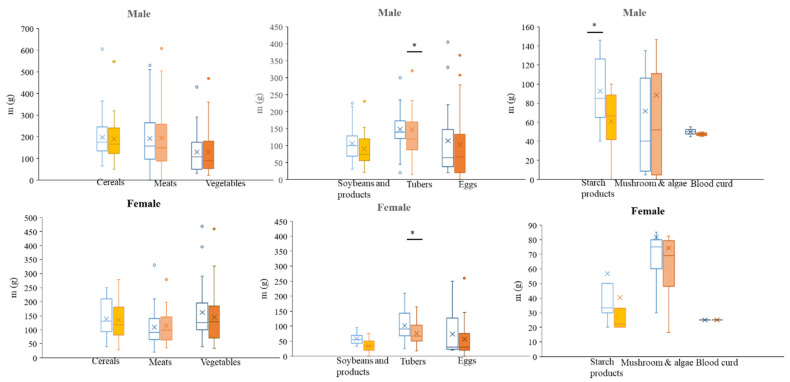
The food intake of different genders according to the weighing and PAD methods (the hollow bar refers to weighing method, and the solid bar refers to PAD), * is the statistical significance, *P* < 0.05.

**Figure 4 nutrients-16-00211-f004:**
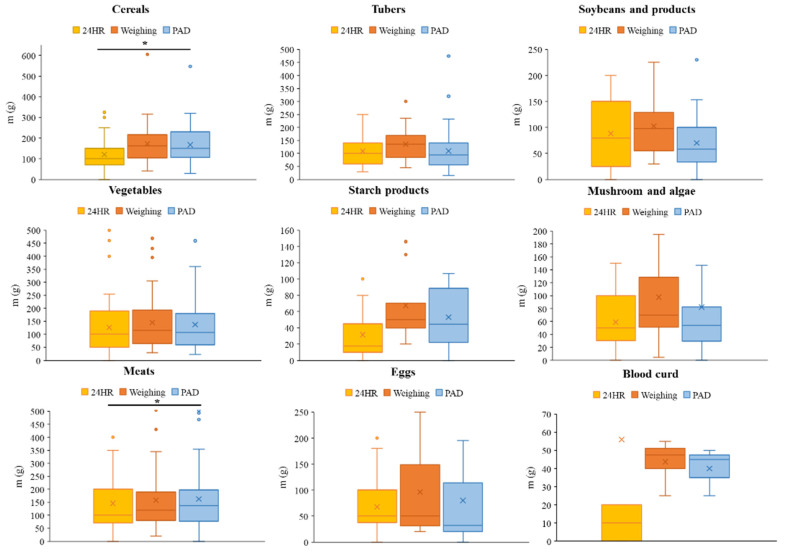
Comparison of food weights estimated using the 24 HR or PAD method with actual food weights, * is the statistical significance, *P* < 0.05.

**Figure 5 nutrients-16-00211-f005:**
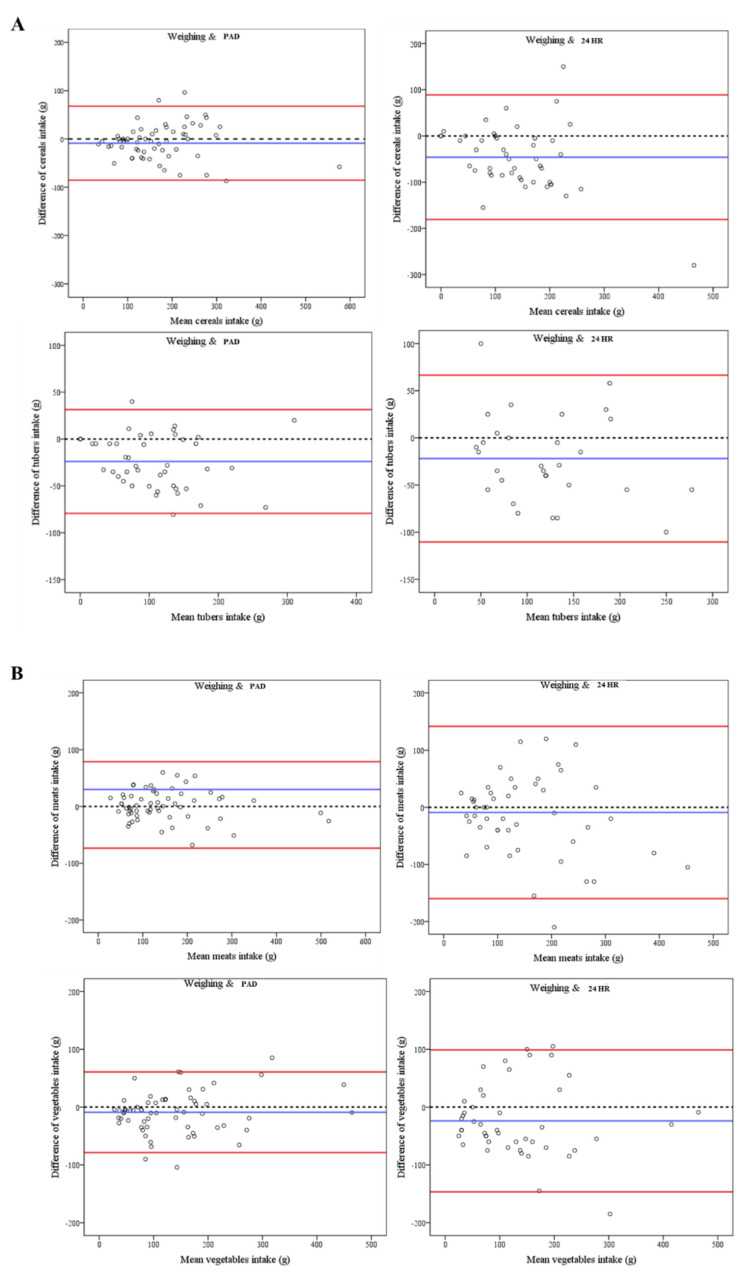
Bland–Altman plots showing the proportional biases of different food intakes between the weighing method and the 24 HR or PAD method. The blue line represents mean differences, and the dashed lines represent 0 values; the red lines represent the 95% limits of agreement (standard deviation: 1.96).((**A**), cereals and tubers estimated with the different methods; (**B**), meats and vegetables estimated with the different methods; (**C**), starch and mushrooms estimated with the different methods; (**D**), soybeans and eggs estimated with the different methods).

**Figure 6 nutrients-16-00211-f006:**
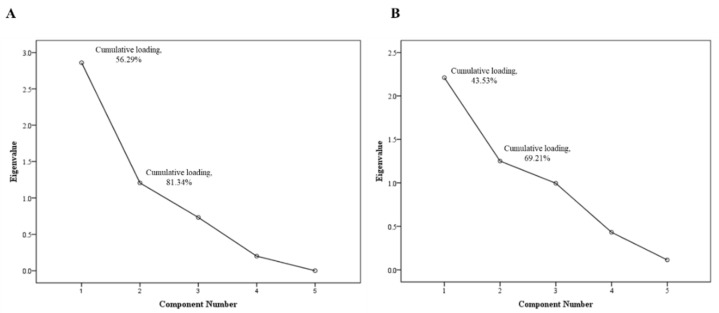
Scree plot of PCA’s bias for estimated weights: (**A**) 24 HR compared to weighing; (**B**) PAD compared to weighing.

**Figure 7 nutrients-16-00211-f007:**
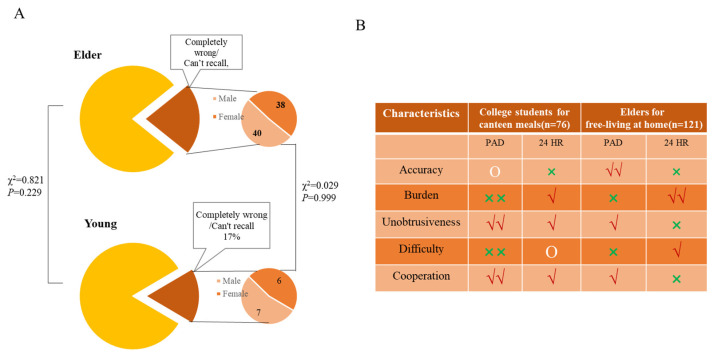
Feedback on the PAD method application among elders and college students: (**A**) Feedback on food records in 24 HR based on the food records in PAD; (**B**) perceptions and attitudes concerning PAD and 24 HR. ×× = very poor; × = poor; O = fair; √ = good; and √√ = excellent.

**Table 1 nutrients-16-00211-t001:** The difference in food mass between weighing and the PAD method for different genders.

Foods	n	Gender	d	D	d%	D%
Cereals	77	Male	−5 (−40, 15)	25 (10, 44.25)	−3.26 (−23.86, 7.69)	13.82 (5.82, 24.52)
		Female	−5 (−22.09, 11.1)	20.46 (5.78, 33.23)	−5.99 (−17.49, 8.51)	15.09 (7.23, 20.71)
		*P*	0.558	0.074	0.868	0.325
Tubers	50	Male	−29.5 (−47.03, 6.25)	31.5 (10.75, 53.825)	−18 (−28.71, 4.44)	21.97 (10, 30.65)
		Female	−30.9 (−47.5, −5)	32.98 (5.15, 47.5)	−30.89 (−40.97, −9.6)	31.69 (13.33, 44.91)
		*P*	0.071	0.213	0.016 *	0.041 *
Soybeans and related products	28	Male	−11.8 (−33.05, 1)	19.5 (5.9, 38.45)	−20.46 (−31.50, 1.79)	27.06 (15.38, 38.91)
		Female	−15 (−21.4, 0)	15 (5.7, 21.4)	−27.06 (−38.91, 0)	0 (0, 0)
		*P*	0.791	0.31	0.884	0.403
Vegetables	108	Male	−5.75 (−20, 21.41)	20 (10, 40.42)	−10 (−20, 18.28)	20 (11.43, 40)
		Female	−10.6 (−41.13, 5.56)	21.21 (8.95, 50.14)	−8.55 (−28.45, 3.42)	14.35 (7.10, 31.96)
		*P*	0.137	0.919	0.369	0.208
Starch products	20	Male	−26.75 (−56.68, −6.43)	26.75 (6.43, 56.68)	−25.08 (−39.02, −15.36)	25.08 (15.36, 39.02)
		Female	−11.1 (−17, −10)	11.1 (10, 17)	−33.33 (−33.33, −28.93)	33.33 (28.93, 33.33)
		*P*	0.403	0.403	0.432	0.432
Mushrooms and algae	25	Male	2.39 (−2.43, 16.03)	5 (3.28, 16.03)	3.85 (−12.65, 20.93)	22.74 (6.65, 42.73)
		Female	−6 (−15.3, 7.5)	13.5 (7.5, 19.43)	−8 (−38, 6.92)	27.82 (8, 38)
		*P*	0.249	0.696	0.33	0.967
Meats	87	Male	−1.2 (−20.24, 22.74)	21.5 (10.33, 37.93)	−0.62 (−12.73, 14.76)	15.5 (5.52, 30.02)
		Female	4.85 (−7.73, 18.33)	15.14 (4.93, 27.02)	2.9 (−5.52, 24.66)	14.29 (5.49, 28.2)
		*P*	0.435	0.013 *	0.31	0.347
Eggs	39	Male	−7.5 (−20.5, 10.35)	17.5 (10, 37.68)	−22.36 (−34.81, 9.68)	24.55 (11.5, 40.71)
		Female	−8.7 (−16.25, 2.88)	11.13 (5, 16.25)	−22.05 (−40, 5.08)	22.36 (7.05, 40)
		*P*	0.621	0.618	0.724	0.844
Blood curd	4	Male	−2.5 (−3.75, −1.25)	2.5 (1.25, 3.75)	−4.55 (−6.82, −2.27)	4.55 (2.27, 6.82)
		Female	0 (0, 0)	0 (0, 0)	0 (0, 0)	0 (0, 0)
		*P*	0.375	0.375	0.375	0.375

PAD: photo-assisted dietary intake assessment; d: relative difference; D: absolute difference; d% = [(estimated weight-actual weight)/actual weight] × 100; D%= [(|estimated weight-actual weight|)/actual weight] × 100. * is the statistical significance, *P* < 0.05.

**Table 2 nutrients-16-00211-t002:** Comparison of differences in food weights estimated using the 24 HR or PAD method with the actual food weights.

Foods	n	Methods	d(g)	D(g)	d%	D%
Cereals	77	24 HR	−50 (−90, 0)	65 (20, 95)	−31.82 (−46.28, 0)	40.59 (14.66, 49.63)
		PAD	−5 (−36.34, 15)	23.42 (10, 44.26)	−4.9 (−24.34, 7.69)	14.28 (5.88, 26.37)
		*P*	0.002 *	0.001 *	0.007 *	0.002 *
Tubers	50	24 HR	−32.5 (−53.75, −1.25)	37.5 (25, 55)	−23.4 (−33.33, −3.7)	27.27 (17.84, 44.27)
		PAD	−24 (−48.75, 3.58)	31.9 (6.93, 50.34)	−20 (32.65, 2.44)	25 (9.09, 42.86)
		*P*	0.503	0.256	0.4	0.784
Soybeans and products	28	24 HR	−7.5 (−30, 28.75)	30 (16.25, 73.75)	−10.1 (−29.82, 0)	28.57 (11.11, 87.5)
		PAD	−5 (−21.53, 0)	15 (0, 24.83)	−2 (−31.5, 0)	0 (0, 0)
		*P*	0.55	0.002 *	0.943	0.002 *
Vegetables	108	24 HR	−35 (−60, 20)	55 (30, 75)	−31.82 (−46.84, 27.5)	44.44 (31.82, 65.22)
		PAD	−5.5 (−20, 15.83)	13.7 (5, 35)	−6 (−19.78, 11.83)	17.67 (9.89, 31.7)
		*P*	0.078	0 *	0.16	0 *
Starch products	20	24 HR	−10 (−66, 10)	50 (10, 66)	−45.21 (−71.43, 0)	50 (25, 92.31)
		PAD	−6.7 (−17, −3.4)	6.7 (3.4, 17)	−16.75 (−34, −4.86)	16.75 (4.86, 34)
		*P*	0.944	0.042 *	0.735	0.063
Mushrooms and algae	25	24 HR	−7.5 (−32.5, −1.2)	20 (5, 42.5)	−25.93 (−61.54, −9.09)	34.85 (11.82, 67.31)
		PAD	4.78 (−2.36, 12.6)	7.5 (4.39, 22.15)	7.31 (−3.1, 16.84)	14.56 (7.11, 37.19)
		*P*	0.279	0.033	0.347	0.049 *
Meats	87	24 HR	−10 (−40, 30)	35 (20, 75)	−4.76 (−33.33, 30.44)	33.33 (18.6, 42.86)
		PAD	4.85 (−11.41, 20.8)	16.54 (7.73, 37.6)	2.9 (−10.67, 21.13)	14.29 (5.49, 28.2)
		*P*	0.103	0 *	0.356	0 *
Eggs	39	24 HR	−10 (−45, 5)	30 (10, 70)	−23.08 (−50, 0)	42.86 (20, 71.43)
		PAD	0 (−15, 9.55)	11.7 (5.6, 32.5)	0 (−40, 8)	22.36 (7.05, 40)
		*P*	0.459	0.003 *	0.696	0.018 *
Blood curd	4	24 HR	−20 (−31.2, 37.5)	37.5 (22.5, 86.25)	−57.78 (−70, 46.97)	57.78 (13.89, 90)
		PAD	0 (−1.25, 0)	0 (0, 1.25)	0 (−2.27, 0)	0 (0, 2.72)
		*P*	0.715	0.068	0.715	0.068

PAD: photo-assisted dietary intake assessment; d: relative difference; D: absolute difference; d% = [(estimated weight-actual weight)/actual weight] × 100; D% = [(|estimated weight-actual weight|)/actual weight] × 100. * is the statistical significance, *P* < 0.05.

**Table 3 nutrients-16-00211-t003:** Spearman correlation coefficients between the actual food weights and the food weights estimated using the 24 HR or PAD method.

Foods	24 HR		PAD	
	r	*P*	r	*P*
Cereals	0.686	<0.001	0.901	<0.001
Tubers	0.776	<0.001	0.743	<0.001
Soybeans and related products	0.394	0.086	0.799	<0.001
Vegetables	0.747	<0.001	0.88	<0.001
Starch products	0.017	0.965	0.836	<0.001
Mushrooms and algae	0.508	0.064	0.885	<0.001
Meats	0.766	<0.001	0.944	<0.001
Eggs	0.708	<0.001	0.755	<0.001
Blood curd	0.4	0.600	0.899	0

**Table 4 nutrients-16-00211-t004:** Bland–Altman analyses of mass estimated via the weighing method and those estimated via the 24 HR or PAD method.

Foods	Weighing and 24 HR			Weighing and PAD		
	Mean Differences (95%Cl)	95%LOA		Mean Differences (95%Cl)	95%LOA	
		Lower	Upper		Lower	Upper
Cereals	−45.96 (−66.15, −25.7)	−180.74	88.83	−8.81 (−18.81, 1.2)	−85.37	67.75
Tubers	−21.97 (−38.53, −5.41)	−110.45	66.52	−23.92 (−32.62, −15.2)	−79.31	31.47
Soybeans and related products	−0.87 (−31.49, 29.75)	−139.67	137.93	−10.43 (−19.96, −0.9)	−53.62	32.76
Starch products	−29.1 (−65.35, 7.15)	−128.43	70.23	−26.87 (−42.77, −12.97)	−72.83	19.09
Vegetables	−23.89 (−42.49, −5.29)	−146.66	98.88	−8.99 (−18.02, 0.04)	−78.66	60.68
Mushroom and algae	−30 (−69.05, 9.05)	−173.63,	113.63	0.29 (−7.41, 7.98)	−32.86	33.44
Meats	−9 (−31.11, 13.11)	−159.9	141.9	2.73 (−6.98, 12.44)	−73.45	78.91
Eggs	−25 (−56, 6)	−178.61	128.61	−11.29 (−24.18, 1.61)	−89.24	66.66

LOA: limits of agreement; 95% Cl: 95% limits of agreement (standard deviation: 1.96).

**Table 5 nutrients-16-00211-t005:** Component matrix of the bias for estimated weights using the 24 HR/PAD method compared with weighing methods.

Methods	Foods	Communalities	Components	
			1	2
PAD	Cereals	0.943	0.835	0.496
	Tubers	0.911	−0.148	0.943
	Meats	0.799	0.893	−0.048
	Vegetables	0.745	0.826	−0.251
	Eggs	0.062	−0.115	0.221
24 HR	Cereals	0.767	0.794	0.370
	Tubers	0.973	0.210	0.964
	Meats	0.869	0.896	−0.257
	Vegetables	0.828	0.869	−0.270
	Eggs	0.631	0.793	−0.039

**Table 6 nutrients-16-00211-t006:** Bias in the 24 HR and PAD methods versus the weighing method.

Bias	Omitting Food	Over-Recall Food	Incorrect Food	D% > 50%
Eggs (n = 37)				
24 HR	1 (33)	1 (33)	0 (33)	4 (33)
PAD	0 (30)	0 (30)	0 (30)	2 (30)
*χ* ^2^	1.182	0.961	-	4.043
*P*	0.822	0.676	-	0.02 *
Tubers (n = 50)				
24 HR	0 (32)	1 (32)	0 (32)	2 (32)
PAD	2 (29)	0 (29)	0 (29)	8 (29)
*χ* ^2^	2.282	0.921	-	5.053
*P*	0.222	1	-	0.037 *
Cereals (n = 77)				
24 HR	0 (41)	1 (41)	0 (41)	8 (41)
PAD	1 (42)	0 (42)	0 (42)	1 (42)
*χ* ^2^	0.988	1.037	-	6.298
*P*	1	0.494	-	0.015 *
Meats with 100% edible parts (n = 43)				
24 HR	0 (26)	3 (26)	2 (26)	7 (26)
PAD	0 (22)	0 (22)	0 (22)	1 (22)
*χ* ^2^	-	2.708	1.766	4.297
*P*	-	0.239	0.493	0.042 *
Meats with less than 100% edible parts (n = 44)				
24 HR	0 (30)	1 (30)	6 (30)	10 (30)
PAD	0 (28)	0 (28)	0 (28)	0 (28)
*χ* ^2^	-	0.95	6.246	11.278
*P*	-	1	0.024	0.001 *
Root and stem vegetables (n = 11)				
24 HR	1 (7)	1 (7)	0 (7)	6 (7)
PAD	0 (7)	0 (7)	0 (7)	1 (7)
*χ* ^2^	1.077	1.077	-	7.143
*P*	0.5	0.5	-	0.029 *
Melon and solanaceous vegetables (n = 41)				
24 HR	0 (22)	0 (22)	0 (22)	8 (22)
PAD	0 (20)	0 (20)	0 (20)	2 (20)
*χ* ^2^	-	-	-	4.014
*P*	-	-	-	0.048 *
Leafy, flower, and sprout vegetables (n = 56)				
24 HR	0 (35)	1 (35)	2 (35)	9 (35)
PAD	2 (31)	0 (31)	0 (31)	3 (31)
*χ* ^2^	2.329	0.899	1.827	4.654
*P*	0.217	1	0.494	0.03 *

Data were expressed as numbers (total numbers); meats with less than 100% edible parts included chicken legs, ribs, and some fish. * is the statistical significance, *P* < 0.05.

## Data Availability

The data presented in this study are available on request from the corresponding author. The data are not publicly available due to privacy considerations.

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
