# Peer review of "The Validity and Feasibility of Utilizing the Photo-Assisted Dietary Intake Assessment among College Students and Elderly Individuals in China"

_nutrients, 2024, doi:10.3390/nu16020211_

Round 1

Reviewer 1 Report

Comments and Suggestions for Authors

 The aim of this study was to develop an accurate and feasible dietary assessment tool, photo-assisted assessing dietary intake (PAD).To assess its accuracy, the PAD food weight was compared with the gold standard weighing method and the 24-hour method. Both the elderly and the college students were enrolled to evaluate the feasibility of PAD.

I don't fully understand the idea of using PAD methods and weighing only for one meal (dinner) and referring to 24 HR in the validation. This should be better explained or presented in the methodology as study design. Secondly, why the feasibility was tested on another, so different group. Additionally, the authors pointed out as a limitation: the assessment of PAD accuracy did not include fruit, but the feasibility of PAD among older adults included fruit. In my opinion, these are serious methodological flaws that do not allow the study to be treated as validation.

The minor comments:

add explanation for :

line 41 - FD; Tables - d,D; explanations for tables and figures

Line 111 - Rather than - collective feding - institutional feeding, canteen meals, communal feeding

line 122 - would be fed a dinner - would receive a dinner

line 262 - genders

Table 6 - "meats" is twice.

The whole manuscript requires careful editorial improvement.

Author Response

Dear reviewer,

We have revised the manuscript in detail as suggested and where appropriate, the comments and suggestions of the reviewers have been incorporated in the revised manuscript. The grammatical errors and inappropriate expression were revised. We hope, with these modifications and improvements based on the reviewers’ comments, the quality of our manuscript would meet the publication standard of Nutrients. We thank the reviewers for their constructive comments, which have resulted in a strengthening of the manuscript. If you have any question, please contact us without hesitation.

Once again, thank you very much for your help to our manuscript processing.

Comment 1:

English language fine. No issues detected.

Response:

Thank you for your comments. We deeply and sincerely appreciate the reviewer’s endorsement.

Comment 2:

Does the introduction provide sufficient? Yes.

Are all the cited references relevant to the research? Yes

Are the conclusions supported by the results? Yes

Response:

Thank you for your comments. We deeply and sincerely appreciate the reviewer’s endorsement.

Comment 3:

Is the research design appropriate? Must be improved

Response:

We express our gratitude for your valuable comments. The study design has been revised to incorporate additional details and align with the intended design, which were marked with the red colour.

Comment 4:

Are the methods adequately described? Must be improved

Response:

Thank you for your comments and suggestions. The methodology section encompassed the operational procedure of the food atlas, the utilization of bowls, dietary menu, and additional details presented in the supplementary materials.

Comment 5:

Are the results clearly presented? Must be improved

Response:

Thank you for your comments and suggestions. We’re very sorry for the confusion expression. The results revised some writing errors including the data analysis from the tables and figures.

Comment 6:

I don't fully understand the idea of using PAD methods and weighing only for one meal (dinner) and referring to 24 HR in the validation. This should be better explained or presented in the methodology as study design.

Response:

Thank you for your valuable comments. We concur with your comments.

  1. In order to assess the validity of our findings, it is imperative to take into account three meals, namely breakfast, lunch, and dinner, within a single day. However, we encountered certain challenges in implementing this approach among the students at this university. Initially, we conducted an investigation pertaining to dining-related information, encompassing dining schedules, preferences, and the composition of meals offered at each canteen. The questionnaire results indicated that a significant majority of students possessed ample time for dinner to cooperate with the current operations. 

In the context of objectivity, this university offers four canteens that provide a diverse range of food options. Students exhibit a tendency to select different canteen stalls for various meal times. However, when considering the feasibility and precision of conducting a dietary survey, it becomes necessary to restrict the selection of canteen stalls for operational purposes. This rule may potentially diminish students' cooperation and satisfaction. Consequently, in order to account for students' dining preferences and practicality, the evaluation of accuracy is limited to the only dinner mealtime. Furthermore, dinners typically serve as the primary meal for the majority of Chinese families, thus making the selection of dinner more aligned with China's national context. Third, prior to the experimental design, the literature was reviewed, and only one meal was also selected to evaluate the validation [1-3].

   Consequently, in this present study, the choice of dinner was made after careful consideration of various factors.

  1. There are several reasons for choosing 24-hour dietary recall (24HR) as the validation method. Firstly, 24HR is extensively employed in dietary surveys due to its widespread utilization. Additionally, the weighing method employed is considered the gold standard for dietary assessment. Meanwhile, 24 HR method with a different error structure from PAD was selected to examining relative validity.

In order to validate the accuracy of the PAD method, two aspects were considered. On one hand, the PAD method aimed to obtain food mass measurements that closely resembled the actual weights of the food. On the other hand, the estimated food mass obtained through the PAD method was expected to be more precise compared to the commonly used 24HR method. The comparative analysis of these aspects suggests that the PAD method demonstrates practicality and generalizability. In addition, the selection of weighting and 24-hour dietary recall (24HR) as reference methods was consistent with findings from other studies [1-6].

In conclusion, only dinner was selected to evaluate the validation of PAD method by referring to the weighing method and comparing with 24 HR.

References:

[1]Ding Y, Lu X, Xie Z, Jiang T, Song C, Wang Z. Evaluation of a Novel WeChat Applet for Image-Based Dietary Assessment among Pregnant Women in China. Nutrients. 2021, 13(9),3158

[2]Six BL, Schap TE, Zhu FM, Mariappan A, Bosch M, Delp EJ, Ebert DS, Kerr DA, Boushey CJ. Evidence-based development of a mobile telephone food record. J Am Diet Assoc. 2010,110(1):74-9

[3]Nicklas TA, O'Neil CE, Stuff JE, Hughes SO, Liu Y. Characterizing dinner meals served and consumed by low-income preschool children. Child Obes. 2012 Dec;8(6):561-71

[4]Folson GK, Bannerman B, Atadze V, Ador G, Kolt B, McCloskey P, Gangupantulu R, Arrieta A, Braga BC, Arsenault J, Kehs A, Doyle F, Tran LM, Hoang NT, Hughes D, Nguyen PH, Gelli A. Validation of Mobile Artificial Intelligence Technology-Assisted Dietary Assessment Tool Against Weighed Records and 24-Hour Recall in Adolescent Females in Ghana. J Nutr. 2023,153(8):2328-2338.

[5]Whitton C, Healy JD, Collins CE, Mullan B, Rollo ME, Dhaliwal SS, Norman R, Boushey CJ, Delp EJ, Zhu F, McCaffrey TA, Kirkpatrick SI, Atyeo P, Mukhtar SA, Wright JL, Ramos-García C, Pollard CM, Kerr DA. Accuracy and Cost-effectiveness of Technology-Assisted Dietary Assessment Comparing the Automated Self-administered Dietary Assessment Tool, Intake24, and an Image-Assisted Mobile Food Record 24-Hour Recall Relative to Observed Intake: Protocol for a Randomized Crossover Feeding Study. JMIR Res Protoc. 2021, 10(12):e32891. 

[6]Nguyen PH, Tran LM, Hoang NT, Trương DTT, Tran THT, Huynh PN, Koch B, McCloskey P, Gangupantulu R, Folson G, Bannerman B, Arrieta A, Braga BC, Arsenault J, Kehs A, Doyle F, Hughes D, Gelli A. Relative validity of a mobile AI-technology-assisted dietary assessment in adolescent females in Vietnam. Am J Clin Nutr. 2022,116(4):992-1001. 

Comment 7:

Secondly, why the feasibility was tested on another, so different group.

Response:

Thank you for your valuable comments and suggestions. I would like to apologize for any confusion caused by my previous expression. In order to comprehensively assess the viability and implementation of the dietary survey, it is necessary to consider various age groups and diverse meal patterns, encompassing both communal dining facilities and home-cooked meals. This approach ensures that the majority of dining styles are accounted for. Additionally, the survey should be conducted specifically for the target population, namely college students lacking life and cooking experience, as well as elderly individuals with declining physiological functions. These groups require special attention due to their susceptibility to reduced accuracy in reporting dietary information. The enrollment of college students was conducted to validate the effectiveness of PAD, while both college students and elders were included to assess the feasibility of PAD. The college students constituted the same population in both evaluations, with the exception of differences in enrollment numbers. Specifically, 76 students were enrolled for the feasibility evaluation, while 71 students were enrolled for the validation evaluation, with five students being excluded due to incomplete or inaccurate weighing data. Additional explanations were provided in the methodology section of the revised manuscript.

Comment 8:

Additionally, the authors pointed out as a limitation: the assessment of PAD accuracy did not include fruit, but the feasibility of PAD among older adults included fruit. In my opinion, these are serious methodological flaws that do not allow the study to be treated as validation.

Response:

We express our gratitude for the highly valuable suggestions provided. In fact, fruits intake was assessed in this current study both in the feasibility evaluation. Our findings indicate that fruits are not typically consumed alongside meals, but rather during leisure periods such as class breaks, after engaging in physical activities, and during mid-day naps. In collage students’ feedback, they are customary to consume fruits either in their entirety or in halves, rather than quantifying them in containers prior to consumption. Furthermore, it is imperative to adhere to the prohibition of serving raw food ingredients, including edible raw vegetables and fruits, in group feeding canteens, as mandated for the purpose of ensuring food safety during the CoVID-19 period. Therefore, the fruit weights estimation without PAD was excluded in the validation analysis.

Comment 9:

add explanation for :

line 41 - FD; Tables - d,D; explanations for tables and figures

Response:

Thank you for your comments and suggestions. I’m very sorry for my carelessness. FD, a writing error, it's actually written as DR, short for dietary record. I checked the whole manuscript carefully, and revised the errors including writing and grammatical errors.  Additionally, comprehensive details and explanatory notes have been incorporated for all tables and figures. 

Comment 10:

Line 111 - Rather than - collective feding - institutional feeding, canteen meals, communal feeding

line 122 - would be fed a dinner - would receive a dinner

Response:

Thank you for your suggestions. I apologized for the inappropriate expressions. The words were correct as canteen meals in the whole manuscript including figure 6, and other inappropriate expressions and grammatical errors were revised.

Comment 11:

line 262 – genders

Response:

Thank you for your comments and suggestions. I’m very sorry for my carelessness. The word was revised.

Comment 12:

Table 6 - "meats" is twice.

Response:

 I sincerely apologize for any confusion caused by the previous expressions. In order to assess the precision of PAD in estimating the weight of meats, we have categorized the meat into two distinct groups: those comprising 100% edible parts and those containing less than 100% edible parts, such as chicken legs, ribs, and fish. Consequently, we have revised the terminology to refer to these groups as "meat with 100% edible parts" and "meats with less than 100% edible parts."

Reviewer 2 Report

Comments and Suggestions for Authors

Fan et al. conducted a study on "Validity and feasibility of photo-assisted assessing dietary in-take for dietary assessment among college students and elders 3 in China." The paper has some merits but is poorly presented in its current form for the following reasons. First, comprehending several aspects of the manuscript is very hard. It is a promising paper, but I will suggest an English language editor to revise the paper thoroughly. Second, the question as to who and how the meals were prepared was not clearly articulated in the methods. Also, a huge setback of the study is the curated specially designed bowls and dishes cooked based on the menu in a buffet style, which may not be representative of the population. This calls to question the viability of the PAD models. The statistical methods were not well described, particularly considering the equations in lines 221-230. For example, the relative and absolute weight equations appear similar and further clarifications would be necessary. Also, the statement “multiple linear regressions between AS…” in the paper is vague. What is the meaning of this? Critical details of methods and the underlying basis of the statistical analysis are missing in the paper. Third, the presentation of findings in the results is quite unreflective of the tables and figures presented. Itemizing the findings without highlighting critical statistics from the tables and figures is hardly appropriate. However, there was no description of the characteristics of respondents in this study. These details and many others would be useful in adequately revising the paper for scientific soundness.

Comments on the Quality of English Language

Comprehending several aspects of the manuscript is very hard. It is a promising paper, but I will suggest an English language editor to revise the paper thoroughly.

Author Response

Dear reviewer,

Journal: Nutrients

Manuscript ID: nutrients-2741942

Title: Validity and feasibility of photo-assisted assessing dietary intake for dietary assessment among college students and elders in China

We would like to submit the enclosed manuscript entitled “Validity and feasibility of photo-assisted assessing dietary intake for dietary assessment among college students and elders in China” for the publication in “Nutrients”.

We have revised the manuscript in detail as suggested and where appropriate, the comments and suggestions of the reviewers have been incorporated in the revised manuscript. The grammatical errors and inappropriate expression were revised. We hope, with these modifications and improvements based on the reviewers’ comments, the quality of our manuscript would meet the publication standard of Nutrients. We thank the reviewers for their constructive comments, which have resulted in a strengthening of the manuscript. If you have any question, please contact us without hesitation.

Once again, thank you very much for your help to our manuscript processing.

Point by point responses to the reviewers’ comments with blue are listed below this letter.

Reviewer 2

Comment 1:

English very difficult to understand/incomprehensible

Response:

Thank you for your suggestions. I’m very sorry for my grammatical errors and poorly expressions. I revised language thoroughly.

Comment 2:

Does the introduction provide sufficient? Must be improved

Response:

Thank you for your comments. The introduction was revised.

Comment 3:

Are all the cited references relevant to the research? Can be improved

Response:

Thank you for your comments. We revised references including supplemented some references.

Comment 4:

Is the research design appropriate? Can be improved

Response:

We express our gratitude for your valuable comments. The study design has been revised to incorporate additional details and align with the intended design, which were marked with the red colour.

Comment 5:

Are the methods adequately described? Can be improved

Response:

Thank you for your comments. Some information was supplemented including critical details of methods and the underlying basis of the statistical analysis.

Comment 6:

Are the results clearly presented? Can be improved

Response:

Thank you for your comments. The results part was revised marked with the red colour including presentation of findings from tables and figures and participants characteristics provided in the supplementary materials.

Comment 7:

Are the conclusions supported by the results? Can be improved

Response:

Thank you for your suggestions. The conclusions was revised.

Comment 8:

First, comprehending several aspects of the manuscript is very hard. It is a promising paper, but I will suggest an English language editor to revise the paper thoroughly.

Response:

We deeply and sincerely appreciate the reviewer’s endorsement for the content. But, I’m very sorry for my grammatical errors and poorly expressions. I revised language thoroughly.

Comment 9:

Second, the question as to who and how the meals were prepared was not clearly articulated in the methods. Also, a huge setback of the study is the curated specially designed bowls and dishes cooked based on the menu in a buffet style, which may not be representative of the population. This calls to question the viability of the PAD models.

Response:

Thank you for your valuable comments and suggestions. I sincerely apologize for the omission of crucial information. In order to validate the PAD method, the selection of the menu was meticulously determined by adhering to the Chinese dietary guidelines (2016) and by incorporating elements from the previous cafeteria menu, which encompassed 30 food items comprising both dishes and staples. For further details, please refer to the supplementary materials where the menu is provided.

The college canteens in this study prepared cooked foods that were influenced by various food cultures. The selection of buffets not only catered to the need for diverse food options, but also accommodated different eating habits, thereby representing a wider range of regional flavors and dietary preferences.

Comment 10:

The statistical methods were not well described, particularly considering the equations in lines 221-230. For example, the relative and absolute weight equations appear similar and further clarifications would be necessary.

Response:

Thank you for your comments and suggestions. I will apologize the confusion the statistics. First, the relative difference (d) was calculated as d = data derived from the 24 HR or PAD method – data derived from the weighing method.

the absolute difference (D) was calculated as D = |data derived from the  24 HR or PAD method − data derived from the weighing method|.

In fact, the D is the absolute value of d. These value including d, D, d% ,D% was referenced with the previous reports [1,2] .

References:

[1]Ding Y, Lu X, Xie Z, Jiang T, Song C, Wang Z. Evaluation of a Novel WeChat Applet for Image-Based Dietary Assessment among Pregnant Women in China. Nutrients. 2021, 13(9),3158

[2]Ding, Y.; Yang, Y.; Li, F.; Shao, Y.; Sun, Z.; Zhong, C.; Fan, P.; Li, Z.; Zhang, M.; Li, X.; et al. Development and validation of a photographic atlas of food portions for accurate quantification of dietary intakes in China. J. Hum. Nutr. Diet. 2021, 34, 604–615.

Comment 11:

Also, the statement “multiple linear regressions between AS…” in the paper is vague. What is the meaning of this?

Response:

I would like to express my gratitude for your valuable comments and suggestions. I would also like to extend my sincere apologies for employing an incorrect analysis method. The use of multiple linear regression to assess the contribution of individual food groups to the accuracy of the PAD/24HR method is not appropriate. To determine the food item that had a more pronounced impact on the accuracy of the PAD/24HR method, the revision manuscript employed principal component analysis (PCA) [3-5]. The method section of the manuscript provided a comprehensive explanation of the PCA procedure. The outcomes of the PCA analysis indicated that the estimated mass of cereals, tubers, eggs, and meats had a more significant effect on the accuracy of the 24 HR and PAD method.

Reference:

[3]Cacau LT, De Carli E, de Carvalho AM, Lotufo PA, Moreno LA, Bensenor IM, Marchioni DM. Development and Validation of an Index Based on EAT-Lancet Recommendations: The Planetary Health Diet Index. Nutrients. 2021, 13(5):1698.

[4]Santos RO, Gorgulho BM, Castro MA, Fisberg RM, Marchioni DM, Baltar VT. Principal Component Analysis and Factor Analysis: differences and similarities in Nutritional Epidemiology application. Rev Bras Epidemiol. 2019,22:e190041.

[5]Geigl C, Loss J, Leitzmann M, Janssen C. Social Factors of Dietary Risk Behavior in Older German Adults: Results of a Multivariable Analysis. Nutrients. 2022,14(5):1057

Comment 12:

Critical details of methods and the underlying basis of the statistical analysis are missing in the paper.

Response:

I express my gratitude for your valuable suggestions, as they have the potential to enhance the overall quality of my manuscript. The manuscript has been revised to include essential details regarding the methods employed and the fundamental principles underlying the statistical analysis, which are indicated by the use of red font. Additionally, supplementary materials have been included to provide necessary information, such as the menu.

Comment 13:

Third, the presentation of findings in the results is quite unreflective of the tables and figures presented. Itemizing the findings without highlighting critical statistics from the tables and figures is hardly appropriate.

Response:

Thank you for your values suggestions, which can improve my quality of manuscript. The result from figures and tables were revised.

Comment 14:

However, there was no description of the characteristics of respondents in this study.

Response:

Thank you for suggestions. I’m very sorry for missing the important information. The characteristics of participants were provided in the supplementary materials.

Round 2

Reviewer 1 Report

Comments and Suggestions for Authors

Thank you for your explanation and taking into account my comments. In the submitted version, it is necessary to standardize the fonts and adapt them to editing requirements. There should always be text before a table/figure with a reference to it.

Author Response

Comment 1:

English language fine. No issues detected.

Response:

Thank you for your comments. We deeply and sincerely appreciate the reviewer’s endorsement.

Comment 2:

Does the introduction provide sufficient? Yes.

Are all the cited references relevant to the research? Yes

Is the research design appropriate? Yes

Are the methods adequately described? Yes

Are the results clearly presented? Yes

Are the conclusions supported by the results? Yes

Response:

Thank you for your comments. We deeply and sincerely appreciate the reviewer’s endorsement.

Comment 3:

 In the submitted version, it is necessary to standardize the fonts and adapt them to editing requirements. There should always be text before a table/figure with a reference to it.

Response:

We express our gratitude for your valuable comments. The manuscript has been revised including written errors, standardization of format, which were marked with the red colour. In addition, the description of figures and tables were located before figures and tables.

Reviewer 2 Report

Comments and Suggestions for Authors

The authors have addressed all comments.

Comments on the Quality of English Language

The English language presentation has improved significantly. However, there are a few minor edits required to improve the manuscript.

Author Response

Comment 1:

Minor editing of English language required.

The English language presentation has improved significantly. However, there are a few minor edits required to improve the manuscript.

Response:

Thank you for your suggestions. I’m very sorry for my grammatical errors and poorly expressions. I revised language thoroughly.

Comment 2:

Does the introduction provide sufficient? Yes.

Are all the cited references relevant to the research? Yes

Is the research design appropriate? Yes

Are the methods adequately described? Yes

Are the results clearly presented? Yes

Are the conclusions supported by the results? Yes

Response:

Thank you for your comments. We deeply and sincerely appreciate the reviewer’s endorsement.